# A Large-Scale High-Throughput Screen for Modulators of SERCA Activity

**DOI:** 10.3390/biom12121789

**Published:** 2022-11-30

**Authors:** Philip A. Bidwell, Samantha L. Yuen, Ji Li, Kaja Berg, Robyn T. Rebbeck, Courtney C. Aldrich, Osha Roopnarine, Razvan L. Cornea, David D. Thomas

**Affiliations:** 1Department of Medicine, Cardiovascular Division, University of Minnesota, Minneapolis, MN 55455, USA; 2Department of Biochemistry, Molecular Biology and Biophysics, University of Minnesota, Minneapolis, MN 55455, USA; 3Department of Medicinal Chemistry, University of Minnesota, Minneapolis, MN 55455, USA

**Keywords:** calcium ATPase, calcium transport, drug discovery, membrane transport, cardiac muscle, heart failure

## Abstract

The sarco/endoplasmic reticulum Ca-ATPase (SERCA) is a P-type ion pump that transports Ca^2+^ from the cytosol into the endoplasmic/sarcoplasmic reticulum (ER/SR) in most mammalian cells. It is critically important in muscle, facilitating relaxation and enabling subsequent contraction. Increasing SERCA expression or specific activity can alleviate muscle dysfunction, most notably in the heart, and we seek to develop small-molecule drug candidates that activate SERCA. Therefore, we adapted an NADH-coupled assay, measuring Ca-dependent ATPase activity of SERCA, to high-throughput screening (HTS) format, and screened a 46,000-compound library of diverse chemical scaffolds. This HTS platform yielded numerous hits that reproducibly alter SERCA Ca-ATPase activity, with few false positives. The top 19 activating hits were further tested for effects on both Ca-ATPase and Ca^2+^ transport, in both cardiac and skeletal SR. Nearly all hits increased Ca^2+^ uptake in both cardiac and skeletal SR, with some showing isoform specificity. Furthermore, dual analysis of both activities identified compounds with a range of effects on Ca^2+^-uptake and ATPase, which fit into distinct classifications. Further study will be needed to identify which classifications are best suited for therapeutic use. These results reinforce the need for robust secondary assays and criteria for selection of lead compounds, before undergoing HTS on a larger scale.

## 1. Introduction

The sarco/endoplasmic reticulum (SR) Ca-ATPase (SERCA) plays a dominant role in clearing cytosolic Ca^2+^ in most cell types. Through its enzymatic cycling, SERCA transports Ca^2+^ across the ER/SR membrane (uptake activity) by Ca^2+^-dependent hydrolysis of ATP (Ca-ATPase activity) [1,2]. In muscle, this process maintains the Ca^2+^ gradient across the SR membrane, lowering the cytosolic Ca^2+^ concentration, as required for muscle relaxation, and maintaining the SR Ca^2+^ content necessary for Ca^2+^ release through the ryanodine receptor (RyR), as required for muscle contraction. Ca^2+^ also controls other key cellular functions, in both muscle and non-muscle cells, such as signaling, apoptosis, and metabolism [3]. Deficient control of Ca^2+^, often due to insufficient SERCA activity, is implicated in numerous disorders, including heart failure [4,5,6], cancer [7], diabetes [8], and muscular dystrophy [9].

Diminished activity of the cardiac SERCA2a isoform in heart failure (HF), often accompanied by increased phospholamban (PLB)/SERCA ratio [10,11,12] and/or decreased PLB phosphorylation [12,13,14], results in slower and less complete relaxation after each beat, elevated diastolic Ca^2+^, and degraded cardiac function. Partial restoration of SERCA activity has been achieved by gene therapy, significantly improving HF symptoms in animal models [15,16], and SERCA2a gene therapy treatment for human HF patients advanced through phase IIb clinical trials [17], where it showed neutral results due to pre-existing rAAV antibodies [18]. Despite this surprising setback, SERCA activation remains a valid and attractive target for HF therapy.

Dysregulation of the skeletal muscle isoform, SERCA1a, is also implicated in the pathology of numerous severe myopathies such as muscular dystrophy and sarcopenia [19,20,21]. In most of these cases, pathogenesis is fueled by excess cytoplasmic Ca^2+^ (in the absence of neural excitation), due to leak via RyR and/or insufficient SERCA1a activity. In many cases, the result is reduced excitation-contraction coupling due to decreased Ca^2+^ inside the SR (Ca^2+^ load). Recent studies indicate that many skeletal myopathies can be remedied by increasing SERCA1a expression [22,23] or decreasing sarcolipin (SLN) expression [24], both of which result in SERCA1a activation.

Our goal is to accelerate therapeutic discovery for multiple myopathies that are characterized by elevated resting intracellular Ca^2+^ levels ([Ca^2+^]), ER stress, mitochondrial ROS production, excessive levels of phosphorylation, and Ca^2+^-promoted proteolysis. These defects are linked to an intracellular vicious cycle that calls for small-molecule therapeutics, which are targeted to restore normal physiology in the affected muscles [22,25,26,27,28,29]. We seek small molecules that increase SERCA activity in cardiac muscle, but also expect to find compounds relevant to skeletal muscle diseases. We have previously developed several high-throughput screening (HTS) strategies that were focused on fluorescence lifetime measurements of genetically encoded fluorescent biosensors, which were used in primary screens of chemical libraries [30,31,32,33]. In contrast, in the present study, we have developed a primary HTS assay directly targeting SERCA’s enzymatic activity. Our primary screen specifically assesses the skeletal SERCA1a isoform, due to its availability in nearly pure form from skeletal muscle tissue, but high homology of sequence, structure, and function, validated by decades of research, suggest a high likelihood of cross reactivity between isoforms. Indeed, most compounds previously found to activate SERCA1a also activate SERCA2a, with some isoform-specific variation in efficacy and potency. 

Importantly, we have identified *distinct classifications* of compounds based on their chemical and physical properties and their functional effects on ATPase and Ca^2+^ uptake activities of SERCA. Such classifications will be essential regardless of the screening approach, and could facilitate mining of existing screening data, in lieu of de novo primary HTS efforts. Future studies will seek to (1) identify additional activating compounds from this ATPase-based screen or other fluorescent biosensor screens, (2) develop analogues optimized via medicinal chemistry, and (3) explore their physiological effects on muscle tissue and whole animals, to advance toward therapeutics.

## 2. Materials and Methods

### 2.1. Compound Handling

The DIVERSet library, containing 46,000 compounds, was purchased from ChemBridge Corporation (San Diego, CA, USA) and formatted into 384-well mother plates (Beckman Coulter, Brea, CA, USA) using a Biomek FX liquid handler (Beckman Coulter). Assay plate copies (Greiner Bio-One, Kremsmunster, Austria) were subsequently made by dispensing 50 nL of either compound (10 µM final concentration per well) or DMSO from the mother plates into transparent polystyrene 384-well plates using an Echo liquid dispenser (Beckman Coulter). Plates were heat-sealed and stored at −20 °C until use. Dimethyl sulfoxide (DMSO, at matching% *v*/*v*) was loaded in columns 1, 2, 23, and 24 as negative controls.

### 2.2. SR Preparations

For skeletal SR, rabbit muscles were freshly harvested from the back and hind leg of New Zealand White rabbits and purified using a sucrose gradient. The resulting light skeletal SR contains 80% SERCA, as a proportion of total protein content [32]. Cardiac SR vesicles were isolated from fresh porcine left ventricular tissue [34].

### 2.3. HTS NADH-Coupled Enzyme Assay Preparation and Measurements

Functional assays were performed using rabbit light skeletal sarcoplasmic (SR) vesicles [32]. For high-throughput NADH-coupled screening, assay mix containing a final concentration of 50 mM MOPS (pH 7.0), 100 mM KCl, 1 mM EGTA, 0.2 mM NADH, 1 mM phosphoenol pyruvate, 10 IU/mL of pyruvate kinase, 10 IU/mL of lactate dehydrogenase, 1 µM of the calcium ionophore A23187, 4 µg/mL skeletal SR, and added CaCl_2_ to obtain pCa 5.4 was dispensed into 384-well DIVERSet assay plates containing 50 nL of compound or DMSO using a FlexDrop Precision liquid dispenser (Perkin Elmer, Waltham, Massachusetts). Plates were sealed, covered from light, and incubated at room temperature for 20 min. The assay was then started by dispensing MgATP (termed “initiation mix”), to a final concentration of 5 mM in each well (80 µL total assay volume) with the FlexDrop liquid dispenser (Perkin Elmer). The time dependence of absorbance was measured at 340 nm in a SpectraMax Plus384 microplate spectrophotometer (Molecular Devices, Sunnyvale, CA). For follow-up concentration response curves of Hit compounds, 400 nL of either compound or DMSO was dispensed into each well using an Echo liquid dispenser (Beckman Coulter). Testing dose-dependent compound effects on rabbit skeletal SR vesicles was done using the same method, as described above in 384-well plates. Determining compound effects on cardiac SR vesicles was done in a similar method as above, with the exception of 10 µg/mL of cardiac SR incubated in assay mix consisting of a final concentration of 50 mM MOPS (pH 7.0), 100 mM KCl, 1 mM EGTA, 1 µM of the calcium ionophore A23187, and added CaCl_2_ to the desired free calcium values. After a 20 min incubation, the “initiation mix”, containing a final concentration of 0.2 mM NADH, 1 mM phosphoenol pyruvate, 10 IU/mL of pyruvate kinase, 10 IU/mL of lactate dehydrogenase, and 5mM MgATP, was dispensed into each well.

### 2.4. Ca^2+^-Uptake Assays

To determine compound effect on the Ca^2+^-uptake activity of SERCA1a and SERCA2a, we used an oxalate-supported assay [35]. A solution containing 50 mM MOPS (pH 7.0), 100 mM KCl, 30 mg/mL sucrose, 1 mM EGTA, 10 mM potassium oxalate, 2 µM Fluo-4, 30 µg/mL cardiac SR vesicles or 12 μg/mL skeletal SR vesicles, and CaCl_2_ calculated to reach the desired free [Ca^2+^] was dispensed into 384-well black walled, transparent bottomed plates (Greiner Bio-One) containing the tested small molecule and incubated at 21 °C for 20 min while covered and protected from light. To start the reaction, MgATP was added to a final concentration of 5 mM, and the decrease in 485-nm excited fluorescence was monitored at 520 nm for 15 min using a FLIPR Tetra (Molecular Devices, San Jose, CA, USA).

## 3. Results

### 3.1. High-Throughput Screening for SERCA Modulators

To identify SERCA-activating compounds from the 46,000-compound DIVERSet library, we adapted an established NADH-coupled enzyme-linked ATPase activity assay [36] to assess changes in ATPase activity in light skeletal SR microsomes prepared from rabbit hind limb muscle—A preparation enriched in SERCA1a [32]. While our primary goal is to identify cardiac therapeutics, skeletal SR preparations were chosen over cardiac SR for the primary screen due to sample purity and reproducibility, in addition to the expectation of extensive overlap between SERCA1a and SERCA2a activators. Typically, such ATPase assays are performed over a range of Ca^2+^ concentrations to assess in detail the effects of compounds on Ca^2+^ affinity and maximum velocity of SERCA activity. However, to streamline our approach for the high-throughput phase of this study, a single [Ca^2+^] concentration (pCa 5.4; i.e., [Ca^2+^]~4 µM) was chosen, which achieves maximal SERCA activity under basal, unstimulated conditions (V_max_). Assay mixtures were added to pre-plated library compounds on 384-well plates for a final assay concentration of 10 µM compound. In total, 145 plates were analyzed over the course of 6 days.

Results from the screen are represented in Figure 1A, depicted as a percent increase or decrease in activity at V_max_, relative to DMSO controls in each assay plate. The percent change in ATPase activity was normally distributed over the entire library, representing a mixture of both activators and inhibitors, and centered on zero (Figure 1B). The coefficient of variance (CV%) calculated for each plate was typical for the SERCA ATPase assay, and remained largely consistent throughout the screen (Figure 1C). A small but significant decline of assay quality (increase in CV%) was noted for several plates near the end of the screen, probably due to decline in performance of the MultiDrop dispensing cassette. Using the CV data in Figure 1C and thapsigargin as a tool compound, the HTS assay quality factor, Z’ [37], was calculated as Z’ = 0.75 ± 0.06 for ΔV_max_ = 100%, Z’ = 0.51 ± 0.11 for ΔV_max_ = 50%, and Z’ = 0.18 ± 0.19 for ΔV_max_ = 30%. An HTS assay is considered excellent when Z’ > 0.5 and feasible when 0.5 ≥ Z’ > 0 [37].

Defining a Hit as a compound that changes ATPase activity by more than 4SD vs. DMSO controls, 70 activators and 138 inhibitors were identified (Figure 1D). Although inhibitors of SERCA are an interesting area of study, numerous high-affinity and well-characterized inhibitory compounds exist, while SERCA-activating compounds are more elusive. Therefore, the remainder of this study focuses on activators, 19 of which were chosen for additional investigation based on the screen response and commercial availability.

### 3.2. Characterization of Hit Compounds in Skeletal SR ATPase Assay

The initial screen assay was performed with compounds at 10 µM. To better study the selected Hit compounds for their activation potential, we performed ATPase analyses (in triplicate) over a range of Hit concentrations (0.05 to 50 µM), generating concentration-response curves (CRCs) to determine maximal effects and EC_50_ (Figure 2). We used the same Ca^2+^ concentration (pCa 5.4) as in the screening assays, thus focusing on the effects on the maximal velocity of SERCA. Two of the selected Hits yielded no response in follow-up assays, probably due to a mismatch in chemical composition between the library compound and the subsequently ordered compound (DS95401435 and DS41547941). This type of discrepancy is a typical hurdle in the screening process. Each of the other 19 compounds produced a substantial increase in ATPase activity, ranging from 30% to 72% (Table 1 and Appendix A), as expected due to similarity of the CRC assay to the initial screen. About half of the compounds were saturable over the dosing range and had EC_50_ of 5 µM or less (Table 1). These follow-up CRC studies also demonstrate the robustness of the initial assay, which clearly identified reproducible modulators of SERCA activity.

### 3.3. Characterization of Hit Compounds in Skeletal SR Ca^2+^ Uptake Assays

While Ca^2+^-dependent ATPase activity is a direct measure of SERCA activity, coupling of ATP hydrolysis to transport of Ca^2+^ across the SR membrane is essential for the role of SERCA in cellular physiology. Therefore, we next asked whether our ATPase-based HTS assay could successfully identify compounds that also increase Ca^2+^ uptake of SERCA. We adapted a Ca^2+^ transport assay to generate CRCs, which yield the maximal effects (efficacy) and EC_50_ (potency) parameters in rabbit skeletal SR, in a similar fashion to the ATPase CRCs (again, at pCa 5.4; (Table 1). Two compounds had no apparent effect on Ca transport, while compound DS33804556 was inhibitory, despite all three substantially increasing ATPase activities (25–42%). Thus, all three compounds decouple Ca^2+^ uptake from ATPase activity, allowing for ATPase hydrolysis in the absence of Ca^2+^ transport. The remaining 16 compounds substantially increased Ca^2+^ uptake (7–31%), although the percent response was generally smaller than the ATPase response (Table 1), indicating increased Ca^2+^ transport at the expense of a loss in efficiency with respect to ATP hydrolysis.

### 3.4. Characterization of Hit Compounds in Cardiac SR ATPase Assay

While initial screening was performed in skeletal SR enriched with the SERCA1a isoform, high sequence and structural homology with the cardiac SERCA2a isoform predicts a potential for cross reactivity of the Hit compounds. To test this hypothesis, we performed ATPase analyses with cardiac SR microsomes, which are enriched for SERCA2a. As with the skeletal SR samples, all Hit compounds dramatically increased ATPase activity at pCa 5.4, indicating that our Hit compounds activate both 1a and 2a isoforms of SERCA (Table 1). In fact, about a third of compounds generated a far greater response in cardiac SR than in skeletal SR, indicating that our methodology provides a robust and global approach to identifying activating compounds of SERCA in differing isoforms.

### 3.5. Characterization of Hit Compounds in Cardiac SR Ca Uptake Assays

To determine whether the observed increases in SERCA2a ATPase activity translate to Ca^2+^ uptake function, we applied our Ca^2+^ transport assay to test our selected Hit compounds in cardiac SR (Table 1). Indeed, 16 of the 19 compounds yielded a clear increase in Ca^2+^ uptake activity. In fact, eight compounds generated an increase at least 50% greater in cardiac SR than in skeletal SR. Three compounds had either an inhibitory or no effect on Ca^2+^ transport, and two of these were also non-activating compounds in skeletal SR.

### 3.6. Compound Physico-Chemical Characteristics

After cheminformatic analysis, the 19 Hits chemically cluster into seven groups, of which five groups contain multiple examples sharing a common scaffold (see Appendix A). The tertiary amide/amine scaffold found in cluster #1 (Figure 3) is the most abundant scaffold, with six Hits. Compounds in this cluster have potencies ranging from 4 to 40 µM in the ATPase assay, correlating closely with the Ca^2+^ uptake assay. The tertiary amide/amine scaffold is attractive for further development, given its tractable structure-activity relationships (SAR), favorable physicochemical properties that include a low molecular weight (~300 Da) and moderate lipophilicity [calculated partition coefficient (clogP) of ~3–4 and total polar surface area (tPSA) of ~30 Å^2^] as well as lack of any conspicuous structural alerts. In general, compounds with molecular weights < 500 and clogP < 5 are desirable, since higher molecular weight drugs tend to have lower permeability, reducing oral bioavailability and tissue distribution, while highly lipophilic drugs tend to have low metabolic stability, poor solubility (which can also limit oral absorption) and reduced receptor selectivity [38,39].

The piperidinyl amides of cluster #2 are typified by compound **9** (Figure 3) and contain three closely related Hits with good activity in the ATPase and the Ca^2+^ uptake assays. This scaffold is also potentially attractive for further development, but the most active analogs are more lipophilic (~logP 4.5) and possess higher molecular weights (~400 Da) than observed for the tertiary amide/amine scaffold (cluster #1). Additionally, the piperidinyl amides contain a potentially labile ethyl ester moiety.

The diarylmethanes of cluster #3 contain three analogs including two closely related compounds with an appended tertiary amide (compounds **10** and **11**) and a third compound containing an oxadiazole amide isostere (compound **12**). Among these analogs, compound **12** (Figure 3) is the most potent in the Ca^2+^ uptake assay (EC_50_ = 0.6 µM in skeletal SR) but has a unique divergent effect on Ca^2+^ uptake between cardiac and skeletal SR (see Discussion), which is not observed with its cluster analogs (compounds **10** and **11**). The limited number of analogs preclude detailed interpretation of the SAR, but the lipophilicity and activity appear to be directly correlated, such that the higher potency of compound **12** comes at the expense of a relatively high molecular weight (MW = 442 Da) and lipophilicity (logP = 4.8).

Several other chemotypes were also identified (Figure 3), including the benzylamino heterocycles (cluster #4) (hetero)aryl amides (cluster #5) and two compounds with no clear chemical analogs (compounds **18** and **19**). Cluster #4, containing only two compounds, has attractive physicochemical properties (MW ~330 Da; cLogP 3.0–4.4) and good activity (ATPase EC_50_ ~3 µM and Ca^2+^ uptake EC_50_ ~1 µM). On the hand, cluster #5 has moderate activity in the ATPase assay (EC_50_ ~3–30 µM) but displays inhibition or no effect on Ca^2+^ uptake. Consequently, cluster #5 is de-prioritized from further development.

The remaining unique compounds with no closely related analogs possess activity commensurate with many of the active clusters. However, both **18** and **19** (Figure 3) contain three linked (hetero)aromatic rings and thus are relatively flat compared to the other identified compounds. In general, compounds with a high degree of planarity are less desirable for drug development, since high planarity is associated with lower receptor selectivity and reduced solubility [40].

## 4. Discussion

### 4.1. Robustness of HTS Assay

Our overall goal is to identify SERCA-activating compounds for development to treat muscle diseases. Therefore, we adapted and miniaturized our well-established NADH-coupled SERCA activity assay [41] to screen a 46,000-compound library. This target-directed screening approach facilitated a high success rate in identifying SERCA-activating compounds. The use of a direct assay for SERCA enzymatic activity in the primary screen was slower (lower throughput) than our previous protein structure-based screening method [32], but it resulted in a pool of compounds that more reliably produced desired effects on SERCA activity.

### 4.2. Compound Classifications: ATPase vs. Ca Uptake Activities

Under ideal conditions, SERCA transports two Ca^2+^ per molecule of ATP hydrolyzed [2,42,43,44]. Some experimental studies have directly observed this 2:1 coupling ratio [45], whereas others have also observed significantly lower enzymatic efficiencies (≤1:1 Ca^2+^:ATP) [46]. Nevertheless, changes in SERCA activity have typically been presumed to have equivalent, proportional effects on both the Ca^2+^ uptake and ATPase activities. The major known exception is regulation by sarcolipin (SLN), which partially uncouples these activities, allowing hydrolysis of ATP in the absence of Ca^2+^ transport, a function found to be critical for thermogenesis [47]. Through our analysis, we identified several classifications of compounds that differentially affect the Ca^2+^ uptake and ATPase functions of SERCA (detailed below). The critical variations were observed in the Ca^2+^ transport component, as all hit compounds strongly increased ATPase activities; this is presumably a selection bias due to the initial primary screen, which measured ATPase activity. Furthermore, all compounds have significantly larger effects on ATPase activity compared to Ca^2+^ uptake at the highest compound concentrations (>10 µM), but have more comparable and/or variable effects at lower concentrations (Table 1). Several compounds induced a 60% or more (as high as 85%) increase in ATPase activity, and all compounds increased ATPase function more than Ca^2+^ uptake, by a factor ranging from 0.5 to 8.

The ability to drive ATPase function much higher could be a result of an intrinsic thermodynamic threshold of SERCA enzymatic activity and/or an artifact of the assay systems. The rate limiting steps in SERCA’s transport process are a sequence of discrete structural transitions allowing for binding of cytosolic calcium and release into the ER/SR lumen, which physically constrains the maximal speed of Ca uptake. Breaking of a single chemical bond for ATP hydrolysis does not have the same thermodynamic limitations and thus can be driven faster than Ca uptake. Therefore, high compound concentrations can push rates of ATPase activity beyond the physical limits of Ca^2+^ transport. On the other hand, these inefficiencies, while important and interesting, appear to only occur at higher than therapeutically relevant concentrations (ideally < 1 mM). Medicinal chemistry may also be able to identify more efficient derivatives. Further physiological and metabolic testing is needed to determine if such inefficiencies extend to the cellular and organ level.

#### 4.2.1. Linked Activating Compounds

We identified several compounds where the initial increases in the CRCs for ATPase and Ca^2+^ transport activities appear closely matched at lower compound concentrations, which we have termed “Linked Activating Compounds” (Figure 4). The ATPase activity response for these compounds appears to rise in unison with Ca^2+^ uptake, up until the Ca^2+^ uptake effect begins to saturate, and ATPase continues to increase. This class of compounds increases SERCA activity while maintaining a consistent coupling ratio, at least at lower concentrations.

#### 4.2.2. Improved Coupling Compounds

We identified another class of compounds where the CRC response in Ca^2+^ uptake precedes that observed with ATPase function, which we have termed “Improved coupling compounds” (Figure 5). At lower concentrations, these compounds increase SERCA Ca^2+^ uptake through an increase in coupling efficiency rather than just driving overall kinetics of the enzyme. As the concentrations increase with these compounds, the ATPase effect catches and surpasses the Ca^2+^ uptake effect. This additional capacity to enhance the coupling efficiency of SERCA suggests that our baseline system and assay conditions are operating below the theoretical 2:1 Ca^2+^ to ATP ratio.

#### 4.2.3. Uncoupling Compounds

We also identified several compounds classified as uncouplers or uncoupling compounds (Figure 6). Similar to the protein inhibitor SLN, these compounds substantially increase SERCA ATPase function at the expense of Ca^2+^ uptake. The primary rate-limiting components of the SERCA reaction cycle are Ca^2+^ binding and subsequent transport. Thus, uncoupling allows for less constrained ATPase hydrolysis, resulting in higher observed activity rates. Interestingly, all compounds were contained in cluster 5. Compounds **15** and **17** have little to no Ca^2+^ uptake effect, while compound **16** has a sizeable inhibitor effect.

#### 4.2.4. Tissue and Isoform Specificity

Dual analysis of our Hit compounds in cardiac and skeletal SR further reveals unique targeting specificity of SERCA modulation. The primary difference between the tissue preparations is the isoform present, with SERCA2a in cardiac SR and SERCA1a in skeletal SR. These sample preparations also have different SERCA regulatory proteins, with cardiac SR containing PLB and skeletal SR containing SLN and myoregulin [48].

While the relative increases in ATPase activity V_max_ were generally consistent between the two sample preparations, the compound effects on Ca^2+^ transport tended to be higher in cardiac SR compared to skeletal (Table 1). In fact, nine compounds had a >50% increase in Ca^2+^ uptake V_max_ effect in cardiac SR compared to skeletal. In addition, compound 16 (DS33804556) was observed to be an inhibitor of Ca^2+^ uptake in both cardiac and skeletal SR. These substantial similarities are likely due to the high degree of homology between SERCA1a and SERCA2a.

The key exception was compound **12** (DS60405307), which shows a strong uncoupling effect in cardiac SR (decreasing Ca^2+^ uptake activity), with significant activation in Ca^2+^ uptake activity in skeletal SR (Figure 7). In addition, the two compounds in cluster 4 (compounds **13** and **14**) were the only ones with a greater than 2-fold increase in Ca^2+^ uptake in cardiac SR vs. skeletal. These discrepant effects are probably due to a critical structural difference between SERCA isoforms, and/or the presence/absence of a tissue-specific SERCA regulatory partner.

### 4.3. Study Limitations

Our initial primary screen, along with all our follow-up ATPase and Ca^2+^ uptake assays, were performed only at a saturating Ca^2+^ concentration, pCa 5.4 (4 µM). While this facilitated data acquisition, we acknowledge that these compounds may differentially affect SERCA at lower Ca^2+^ concentrations. Future screens at different Ca^2+^ are expected to reveal additional unique classifications of SERCA modulating compounds.

In addition, since our primary screen and Hit selection process utilized ATPase activity in skeletal SR, our Hit compounds are biased toward ATPase activators; this might partially explain the higher maximal effects of compounds on ATPase versus Ca^2+^ uptake. In other words, we may have artificially selected compounds that have some uncoupling effect, thus facilitating the enhanced ATPase function.

### 4.4. Next Steps

While this study represents a substantial screening effort, especially from an academic laboratory, a 46,000-compound screen such as ours only represents an industrial pilot screen, with true HTS efforts sometimes utilizing libraries more than 100 times larger. Thus, we believe it would be premature to identify any lead compounds for a full battery of testing. The major takeaways from this study are not the compounds themselves but the strategic advances we have made toward an effective large-scale screening effort. Our primary goal was to optimize selection criteria for in-depth mining of existing screens and/or performance of a larger screen.

Moving forward, assessment of compound effects at lower [Ca^2+^] (e.g., pCa 6.2; i.e., 630 nM) will add another biochemical classification parameter, facilitating better selection criteria for advancement of compounds for further testing. Full Ca^2+^ response curves as is common in SERCA activity assays would also be informative in understanding Ca^2+^ affinity and cooperativity. However, such in-depth analysis will be better suited for the structure-activity response (SAR) phase of the drug discovery process, done in conjunction with medicinal chemistry in order to tease out more nuanced variations between similar compounds. We will also expand our pool of Hit compounds to be tested, from this screen or others, which will be prioritized based on physico-chemical criteria we have identified. Combinations of compounds from different classifications and scaffolds may yield an enhanced and synergistic response. Importantly, we will test the physiological (in situ, in vivo) effects of compounds to determine whether the in vitro activating effects on SERCA produce enhanced muscle function. Furthermore, we speculate that different compound classifications will differentially affect muscle function. Ideally, these avenues will be explored in parallel, as we discover which compound classifications are better suited as therapeutic leads. These additional experimental endeavors are critical before any lead selection. Above all, we are developing a systematic process for the selection of SERCA-activating compounds that could be advanced towards therapeutics of muscle disorders.

## 5. Conclusions

We have developed a robust enzymatic screening assay and used it to identify a pool of SERCA-activating compounds. These compounds yielded distinct effects, divided into biochemical classifications based on Ca^2+^ uptake and ATPase activities. This diversity is likely to be critical in the selection of SERCA-targeted lead compounds. Compounds that improve coupling (between ATPase activity and Ca^2+^ transport) seem especially promising as therapeutics, as they would increase Ca transport with minimal or even reduced energetic demand, reducing concerns about metabolic insufficiencies of increased Ca^2+^ handling and cardiac function.

## Figures and Tables

**Figure 1 biomolecules-12-01789-f001:**
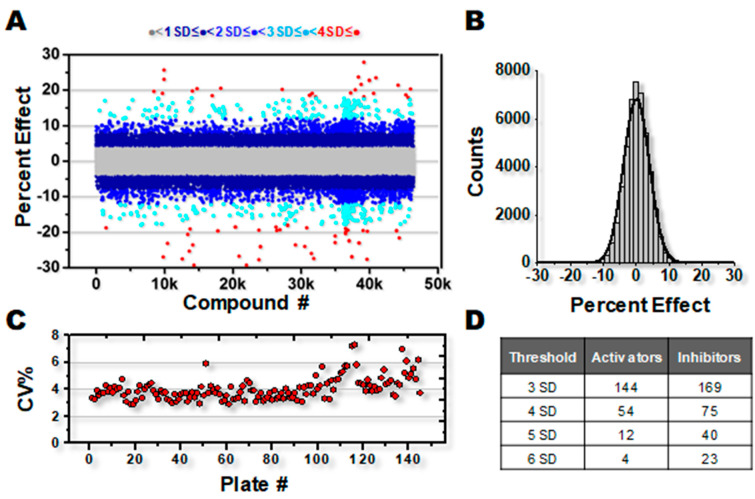
DIVERSet ATPase screen for SERCA effectors. (**A**) Percent change in ATPase activity for 46K compound screen. Selected activating compounds displayed in red. (**B**) Gaussian fit of percent effect distribution. (**C**) CV% for each assay plate. (**D**) Numbers of SERCA activating and inhibiting compounds that meet various SD thresholds above (activators) or below (inhibitors) the mean.

**Figure 2 biomolecules-12-01789-f002:**
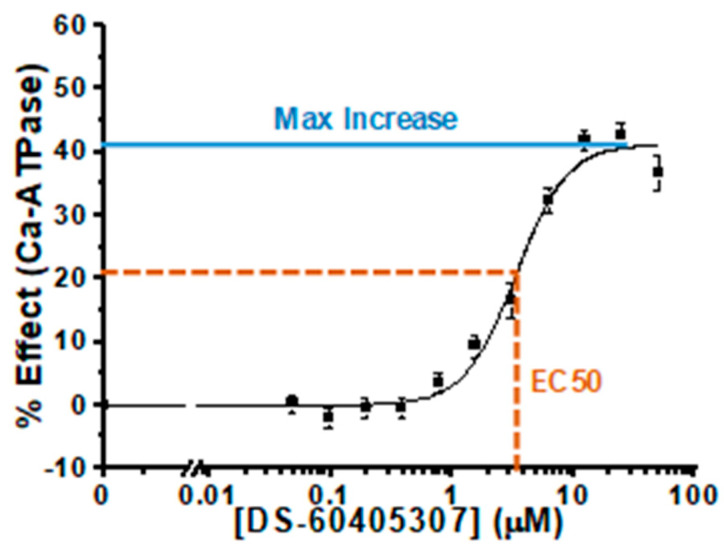
Representative CRC of SERCA-activating compound #12 for ATPase activity assay (% change in activity relative DMSO control) performed at pCa 5.4 in skeletal SR (V_max_).

**Figure 3 biomolecules-12-01789-f003:**
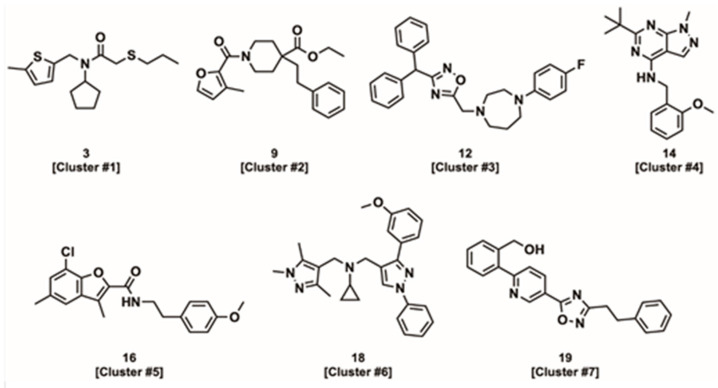
Representative chemical structures from each cluster set.

**Figure 4 biomolecules-12-01789-f004:**
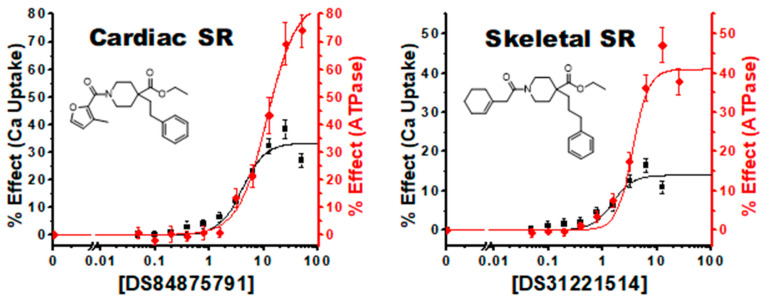
Linked Activating Compounds: Concentration response curves of compounds #9 (DS84875791) in cardiac SR and #7 (DS31221514) display a comparable change at lower compound concentrations, although ATPase achieves a greater maximal effect at the highest compound concentrations. n = 3, ± SEM.

**Figure 5 biomolecules-12-01789-f005:**
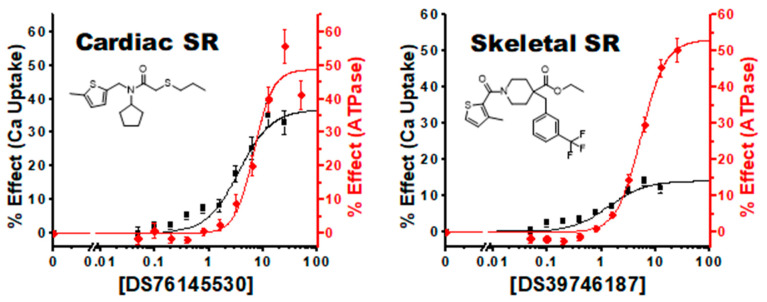
Improved Coupling Compounds: Concentration response curves of compounds #3 (DS76145530) in cardiac SR and #8 (DS39746187) in skeletal SR show a percent increase in Ca^2+^ uptake activity that precedes the ATPase effect indicating improved coupling efficiency between the two SERCA enzymatic activities. n = 3, ±SEM.

**Figure 6 biomolecules-12-01789-f006:**
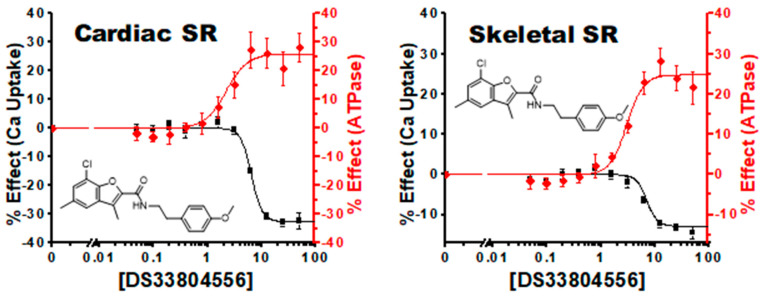
Uncoupling Compounds: Concentration response curve of compound #16 (DS33804556) in cardiac SR and skeletal SR show an inhibition of Ca^2+^ uptake activity despite a substantial increase in ATPase activity.

**Figure 7 biomolecules-12-01789-f007:**
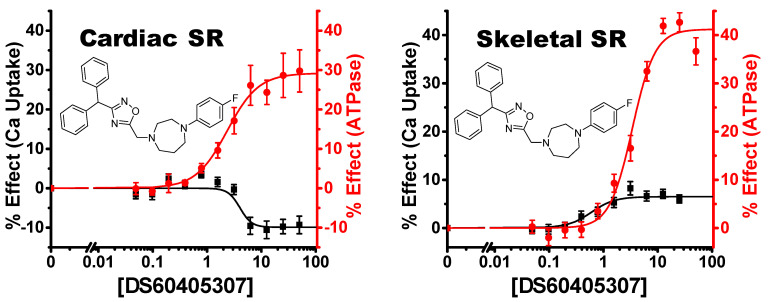
Isoform-specific uncoupler. Concentration response curve of compound #12 (DS60405307) shows an inhibition of Ca^2+^ uptake activity despite a substantial increase in ATPase activity in cardiac SR. However, the same compound has a small but significant increase in Ca^2+^ uptake activity in skeletal SR. This tissue-dependent effect is presumably due to SERCA2a (cardiac) vs. SERCA1a (skeletal) isoform differences or the presence/absence of regulators (PLB vs. SLN). n = 3, ± SEM.

**Table 1 biomolecules-12-01789-t001:** Summary of selected DIVERSet ATPase activators.

		Skeletal SR				Cardiac SR			
Compound		ATPase Activity		Ca^2+^ Uptake Activity		ATPase Activity		Ca^2+^ Uptake Activity	
N, Cluster	DIVERSet ID	Percent Increase	EC_50_	Percent Increase	EC_50_	Percent Increase	EC_50_	Percent Increase	EC_50_
1, 1	67200668	50 ± 25	40 ± 21	27 ± 3	28 ± 9	42 ± 12	31 ± 10	32 ± 2	37 ± 4
2, 1	75082844	36 ± 2	4.0 ± 0.5	15 ± 1	2.2 ± 0.4	33 ± 3	2.9 ± 0.6	12 ± 1	1.2 ± 0.2
3, 1	76145530	54 ± 2	7.3 ± 0.5	20 ± 1	3.4 ± 0.5	48 ± 4	6.7 ± 1.0	37 ± 3	3.5 ± 0.7
4, 1	81260867	53 ± 2	5.4 ± 0.4	14 ± 2	1.4 ± 0.5	58 ± 4	4.8 ± 0.6	24 ± 1	1.6 ± 0.3
5, 1	11966966	41 ± 3	9.0 ± 1.0	10 ± 1	1.4 ± 0.6	58 ± 6	7.7 ± 2	16 ± 3	1.0 ± 0.7
6, 1	25365366	45 ± 8	24 ± 6	9 ± 1	4 ± 1	63 ± 17	29 ± 12	17 ± 1	3.8 ± 0.6
7, 2	31221514	41 ± 3	3.6 ± 0.4	14 ± 2	1.4 ± 0.4	51 ± 3	2.9 ± 0.3	24 ± 3	1.3 ± 0.4
8, 2	39746187	45 ± 2	4.0 ± 0.4	13 ± 1	1.3 ± 0.4	43 ± 2	3.4 ± 0.3	18 ± 2	0.8 ± 0.3
9, 2	84875791	72 ± 2	13.3 ± 0.7	31 ± 7	8 ± 5	85 ± 5	11 ± 1	33 ± 3	4.0 ± 0.8
10, 3	19784159	57 ± 3	10 ± 1	11 ± 3	4 ± 2	55 ± 2	7.5 ± 0.4	16 ± 3	1.6 ± 0.7
11, 3	27035959	61 ± 2	7.2 ± 0.4	9.1 ± 0.8	1.2 ± 0.3	62 ± 2	6.4 ± 0.4	18 ± 2	1.2 ± 0.4
12, 3	60405307	41 ± 2	3.4 ± 0.4	6.5 ± 0.5	0.6 ± 0.2	29 ± 1	2.3 ± 0.3	−10 ± 3	1.3 ± 0.1
13, 4	71721828	36 ± 4	3.2 ± 0.6	8 ± 2	1.1 ± 0.6	36 ± 4	2.9 ± 0.6	21 ± 10	1.8 ± 2
14, 4	32014423	25 ± 3	3.3 ± 0.6	8 ± 2	1.1 ± 0.6	56 ± 3	5.2 ± 0.5	28 ± 5	2.5 ± 0.9
15, 5	19396790	30 ± 1	30 ± 12	--	--	46 ± 6	16 ± 3	6 ± 5	5 ± 11
16, 5	33804556	25 ± 1	3.0 ± 0.3	−10 ± 2	7 ± 13	26 ± 2	2.2 ± 0.4	−27 ± 3	8.9 ± 2
17, 5	39779602	42 ± 4	16 ± 3	--	--	62 ± 5	9.4 ± 1	--	--
18, 6	81801810	39 ± 1	5.2 ± 0.3	8.9 ± 0.1	1.1 ± 0.3	37 ± 1	4.3 ± 0.3	12 ± 1	0.9 ± 0.2
19, 7	30616130	42 ± 2	7.3 ± 0.8	7.2 ± 0.1	1.8 ± 0.7	22 ± 2	6.3 ± 0.4	8 ± 1	2.4 ± 0.6

Compounds are listed 1–19, along with corresponding physicochemical cluster number (see Appendix A) and DIVERSet ID #, along with mean ± SD of percent increase and EC_50_ of a compound for ATPase and Ca^2+^ uptake activities for both skeletal and cardiac SR preparations (n = 3). Dashed entries indicate no observable effect. All CRCs for data in this table are included in Appendix A.

## Data Availability

All data discussed are presented within the article.

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
