# Peer review of "A Large-Scale High-Throughput Screen for Modulators of SERCA Activity"

_biomolecules, 2022, doi:10.3390/biom12121789_

Round 1

Reviewer 1 Report

Bidwell and colleagues developed a high-throughput screening assay targeting the Ca2+-dependent ATPase activity of sarco(endo)plasmic reticulum Ca2+-ATPase (SERCA), and employed this assay to identify SERCA-activating compounds. The selected hit compounds were tested on skeletal and cardiac sarcoplasmic reticulum (SR) vesicles, enriched in SERCA1a and SERCA2a isoform, respectively. The effects of the hit compounds on Ca2+ uptake and ATPase activities were characterized providing distinct compound classifications.

This is a well-written, interesting manuscript that deals with the important research topic of SERCA modulators, particularly SERCA-activating compounds with therapeutic potential for the treatment of skeletal and cardiac muscle disorders.

I have some minor comments for revision or clarification:

1.      The acronym PLB should be defined in the Introduction.

2.      In the abstract the authors mentioned top 22 activating hits, but they actually characterized 19 selected hit compounds, as reported in Table 1.

3.      On page 6, line 217: “The remaining 19 compounds substantially increased Ca2+ uptake…”. As reported in Table 1, 16 compounds significantly increased Ca2+ uptake activity. Same comment on line 236: 16 (not 18) of the 19 compounds yielded a clear increase in Ca2+ uptake activity.

4.      On page 8, line 319: The authors mentioned an intrinsic thermodynamic threshold of SERCA enzymatic activity. Can the authors better explain this concept?

5.      The legend to Figure 4 should be corrected.

Reviewer 2 Report

This manuscript by Bidwell et al. reports new hit matter for small-molecule SERCA activators discovered from a high-throughput screening campaign. The authors adapted a well-established enzyme-coupled NADH-linked ATPase assay and scaled it up to a 384-well format. The authors then used this assay to screen for hit matter using a library of 46,000 compounds; for these assays, the authors used SERCA from cardiac and skeletal muscle. The hits discovered in this assay were further tested using an orthogonal assay to measure calcium uptake in SR microsomal fractions. The complementary screening assays yielded molecules that stimulate calcium transport and SERCA uncouplers, i.e., molecules that stimulate ATPase activity but not active calcium transport. Overall, the study is well-designed and executed. While the overall approach is not innovative as it simply involves scaling up established biochemical assays, the discovery of new hit matter is considered a strength of this study. 

Specific comments:

1. The primary data reported in this study are the % of activation and EC50 values derived from the measurement of specific activity of SERCA. However, It would be informative to complement these studies with full [calcium]-dependent curves to determine the effects of these hits on the relative calcium affinity and cooperativity. 

2. The data presented in this study show a large gap between the increase in ATPase activity and calcium transport. For example, Figure 4 shows n 80% increase in the activity of cardiac SERCA but only a 30% increase in calcium uptake. These findings show that the molecules are inefficient activators, indicating a low 'return on investment' in terms of ATP expenditure. Perhaps the authors could hypothesize on the mechanism(s) underlying the inefficient activation produced by these hits. 

3. It is not clear if the performance of the secondary screening is comparable to that of a medium throughput assay format (e.g., 96-well plate format). It is also not clear how this secondary screening compares to more robust cell-based assays to measure calcium uptake. 

4. A secondary screening assay does not necessarily rely on rapid measurements achieved with fast calcium sensors, so it is not clear why a non-ratiometric dye (Fluo-4) was used instead of a ratiometric one (e.g., Fura red or Fura-2).

Round 2

Reviewer 2 Report

My critiques have been addressed adequately.